# Language bias in orthodontic systematic reviews: A meta-epidemiological study

**Samer Mheissen**[1]*, **Loukia M. Spineli**[2], **Baraa Daraqel**[3,4], **Ahmad Saleem Alsafadi**[5]

**1** Specialist Orthodontist in Private Practice, Syria- Damascus, Syria, **2** Principal Investigator in Evidence Synthesis, Midwifery Research and Education Unit, Hannover Medical School, Hannover, Germany, **3** Department of Orthodontics, Stomatological Hospital of Chongqing Medical University, Chongqing, China, **4** Oral Health Research and Promotion Unit, Al-Quds University, Jerusalem, Palestine, **5** Private Practice, Damascus, Syria

* Mheissen@yahoo.com

## Abstract

### Background

Orthodontic systematic reviews (SRs) include studies published mostly in English than non-English languages. Including only English studies in SRs may result in a language bias. This meta-epidemiological study aimed to evaluate the language bias impact on orthodontic SRs.

### Data source

SRs published in high-impact orthodontic journals between 2017 and 2021 were retrieved through an electronic search of PubMed in June 2022. Additionally, Cochrane oral health group was searched for orthodontic systematic reviews published in the same period.

### Data collection and analysis

Study selection and data extraction were performed by two authors. Multivariable logistic regression was implemented to explore the association of including non-English studies with the SRs characteristics. For the meta-epidemiological analysis, one meta-analysis from each SRs with at least three trials, including one non-English trial was extracted. The average difference in SMD was obtained using a random-effects meta-analysis.

### Results

174 SRs were included in this study. Almost one-quarter ($n = 45/174$, 26%) of these SRs included at least one non-English study. The association between SRs characteristics and including non-English studies was not statistically significant except for the restriction on language: the odds of including non-English studies reduced by 89% in SRs with a language restriction (OR: 0.11, 95%CI: 0.01 0.55, P< 0.01). Out of the sample, only fourteen meta-analyses were included in the meta-epidemiological analysis. The meta-epidemiological analysis revealed that non-English studies tended to overestimate the summary SMD by approximately 0.30, but this was not statistically significant when random-effects model was employed due to substantial statistical heterogeneity (ΔSMD = -0.29, 95%CI: -0.63 to 0.05,

**Data Availability Statement:** Data Availability: The data underlying this paper were shared on open repository: https://doi.org/10.5281/zenodo.10201400.

**Funding:** The author(s) received no specific funding for this work.

**Competing interests:** The authors have declared that no competing interests exist.

P = 0.37). As such, the overestimation of meta-analysis results by including non-English studies was statistically non-significant.

## Conclusion

Language bias has non-negligible impact on the results of orthodontic SRs. Orthodontic systematic reviews should abstain from language restrictions and use sensitivity analysis to assess the impact of language on the conclusions, as non-English studies may have a lower quality.

## Introduction

Systematic reviews (SRs) of randomised controlled trials provide the best summary of evidence to fill the gap by answering a research question, thus, drawing evidence-based practice. Systematic reviews are useful when they have a high methodological quality and include studies with a low risk of bias in the conduct, analysis and reporting [1]. Several shortcomings of SRs were reported in the orthodontic literature. For instance, the lack of grey literature searches [2], the language restriction in the search [2], lack of prediction interval reporting [3], high statistical heterogeneity in the meta-analysis [4], meta-analysis results distorted by small-study effects and publication bias [5], and flaws in the reporting and interpretation of their abstracts [6]. The limitations mentioned above may lead to low-quality evidence [7], as SRs should use standardised and transparent methods to reduce bias and introduce reliable evidence [8].

A well-conducted SR should identify all relevant studies to the question at hand. For this purpose, the Cochrane handbook recommends extensive search in multiple bibliographic databases enhanced by manual and grey literature searches without restriction on the language or the search date [9]. The sensitivity of the search is considered high when the search identifies approximately all relevant reports. Increasing the sensitivity of the search will decrease its precision, as the search will retrieve a large number of studies to include mostly the relevant studies as much as possible. The search precision is defined as the ratio of relevant reports to all identified reports [9]. For instance, if the number of included studies was ten and the search resulted in 1000 records, the precision would be 0.01 (10/1000). A high-standard search strategy guarantees the balance between the search's sensitivity and precision [9].

The English language is considered the dominant language in research [10] including dentistry. While publications from languages other than English is usually considered as of secondary importance. A previous study [11] found that German authors are more likely to publish trials in English when results are statistically significant, increasing the risk of language bias. Likewise, authors from less developed countries tend to publish more positive results than negative results [12]. Language bias may result from publishing significant findings in English language more than other languages [13]. Subsequently, results from only English language studies could provide a biased assessment of the topic.

Systematic review authors usually include studies published in English or languages spoken by the investigators' team. The search restriction to studies written in the English language was found in approximately 70% of orthodontic SRs [2], which may lead to language bias [11]. A meta-epidemiological study [14] indicated that including non-English language studies might not change the results of some SRs. However, not posing language restrictions to the retrieved studies may enable the researchers to inspect possible geographical bias in reporting positive results [15] and improve the quality of the conclusions drawn from the SRs [2].

A recent review [15] included methodological studies investigating the impact of the English language restriction on the effect estimate and found that the search restriction to English studies only had little impact on the effect estimate. To our knowledge, it is unknown if language bias may impact the meta-analysis results in orthodontics. Therefore, this study aimed to explore whether including non-English language studies in orthodontic SRs may affect the meta-analysis results and to estimate the extent of language bias in orthodontic SRs.

## Materials and methods

### Protocol and registration

The reporting of this study followed the proposed items to be used for reporting meta-epidemiological methodology research [16]. There was no registration for the protocol.

### Eligibility criteria

This study included orthodontic SRs published between 1 January 2017 and 31 December 2021 in five orthodontic journals with the highest impact factor (2021): American Journal of Orthodontic and Dentofacial Orthopedics (3.6), European Journal of Orthodontics (4.3), Progress in Orthodontics (3.7), Angle Orthodontist (3.4), and Orthodontics & Craniofacial Research (2.8). Also, Cochrane orthodontic reviews in the same period were included. Methodological studies, scoping reviews, literature reviews, and systematic reviews with fewer than two included studies were excluded.

### Search and study selection

An electronic search was undertaken in Medline via PubMed and Cochrane library for systematic reviews on 20 June 2022. One author (SM) performed the search of PubMed using text words and medical subject headings to retrieve systematic reviews published in the leading orthodontic journals indexed in PubMed (Table 1). All relevant Cochrane orthodontic reviews within the same period were also retrieved by another author (BD) through the Cochrane Oral Health Group. Initial screening for titles and abstracts was performed independently and duplicated by two authors (ASA and BD). Furthermore, two authors (ASA and BD) scrutinised the full text of the potential articles for eligibility. In the presence of disagreement, a consensus was reached after a discussion with a third author (SM).

### Data collection process

A pilot assessment of 30 SRs was undertaken between two authors (BD and ASA) to ensure consistency in the data extraction. After reaching 100% agreement, the same two authors

**Table 1. The search strategy in PubMed via medline.**

| Search Number | Query | Filters | Results |
|---|---|---|---|
| 1 | "Malocclusion"[Mesh] OR "Malocclusion, Angle Class III"[Mesh] OR "Malocclusion, Angle Class II"[Mesh] OR "Malocclusion, Angle Class I"[Mesh] OR "Malocclusion and Short Stature" [Supplementary Concept] | | 34,803 |
| 2 | orthodontic OR orthodonti* | | 90,239 |
| 3 | ("Orthodontics"[Mesh]) OR ("Orthodontics, Preventive"[Mesh] OR "Orthodontics, Interceptive"[Mesh] OR "Orthodontics, Corrective"[Mesh]) | | 54,829 |
| 4 | #1 OR #2 OR #3 | | 102,848 |
| 5 | #4 | Systematic Review | 1,878 |
| 6 | #5 | Systematic Review, from 2017–2021 | 1,156 |

extracted the data and a third author (SM) cross-checked the collected data. All data were entered in a pre-pilot Microsoft Excel® (Microsoft, Redmond, Washington, USA). The following characteristics for each SR were extracted: the number of authors, continent of the first author, publication year, review type (Cochrane and non-Cochrane), protocol registration, the number of included studies, including non-English studies (yes, no), whether studies were excluded based on the language (yes, no), the number of non-English languages, language restriction in the inclusion/ exclusion criteria (yes, no), involvement of librarian in search (yes, no), type of SR (interventional, epidemiological, or diagnostic), and type of included studies (human, animal, or in vitro). We calculated the precision of the search for each SR by dividing the number of included studies by the number of search results after removing duplicates. If the SR included a non-English language, additional information regarding meta-analysis (MA) was extracted: non-English language included in MA (yes, no), and the statistical significance of summary effect estimate (yes, no). One outcome was selected from each SR that included at least one non-English study to investigate the impact of the non-English language on the meta-analysis results. Specifically, we considered the following 'algorithm': if the meta-analysis of primary outcome included at least three studies with at least one being non-English, this meta-analysis was included in our collection. If more meta-analyses were eligible, we opted for the first one addressing the primary outcome. If the meta-analysis of the primary outcome was not eligible, the meta-analysis for secondary outcomes was checked. Therefore, the corresponding forest plot should include at least one non-English study to extract the data. We opted for arm-level data (information reported in each arm of every trial); otherwise, we extracted the data in the contrast-level format as reported in the forest plot.

## Statistical analysis and data synthesis

**Multivariable binary logistic regression.** Proper descriptive statistics were undertaken to summarise the collected characteristics. For categorical characteristics, the absolute and relative frequencies were reported. The median, interquartile range, and minimum and maximum values were provided for metric characteristics. A multivariable binary logistic regression was implemented to explore the association of including a non-English language (yes, no) with each characteristic. All characteristics were included in the model simultaneously, and no variable selection approach (e.g., stepwise selection) was performed. Due to convergence issues stemming from separation in some characteristics, Firth's bias reduction approach was considered to improve estimation [17]. The results were presented in odds ratio (OR), 95% confidence intervals (CI), and p-values based on the profile penalised likelihood. We concluded a statistically significant association when the 95% CI did not include value of no association (OR = 1), which coincides with a p-value less than 5%; otherwise, the association was statistically non-significant, which coincides with a p-value at least 5%. These CIs are preferred to the Wald CIs in the presence of separation [18]. Precision is measured as percentages and was transformed into the logit scale to be included in the model. The publication year was centred on its mean value to improve the interpretation and convergence of the regression coefficient.

**Meta-epidemiological analysis and sensitivity analysis.** A two-stage approach was performed to analyse the meta-epidemiological data and estimate the average bias attributed to non-English studies (S1 File). The meta-epidemiological data comprised one meta-analysis from each eligible SR according to the aforementioned algorithm. The standardised mean difference (SMD) was the effect measure since all selected meta-analyses referred to a continuous primary outcome. Initially, a random-effects meta-regression was conducted to estimate the difference in standardised mean difference (ΔSMD) between non-English and English studies in each meta-analysis. Then a random-effects meta-analysis was performed to combine the

ΔSMDs across the meta-analyses. We also pooled ΔSMDs using a fixed-effect meta-analysis as a sensitivity analysis to the model assumptions. A negative ΔSMD would indicate that non-English studies overestimate the SMD. The supplementary material provides further details on the meta-epidemiological analysis.

All analyses were conducted in the statistical software R (version 4.2.0) [19]. The summary statistics table was created using the R package *gtsummary* [20]. We used the R package *logistf* to apply logistic regression with Firth's bias reduction [18], and the R package *metafor* to conduct the random-effects meta-regression using the function *rma.mv* [21]. The bar plots and forest plots were created using the R-package *ggplot2* [22].

## Results

### Systematic review selection

A total of 1168 SRs were initially identified (Fig 1). Of those, 983 SRs were removed for referring to non-relevant journals. After full text reading of one hundred eighty-five SRs, 174 SRs were included in the present study. Of the 11 excluded SRs, four were methodological studies, two were scoping reviews, two were withdrawn, one did not find any eligible studies, one was a clinical guideline, and one was an erratum (S1 Table).

### Characteristics of the systematic reviews

The non-English studies were included in around one quarter (*n* = 45/174, 26%) of the SRs, representing 3.78% of the total included studies in orthodontic SRs (98/2568). Most SRs included four to six authors (*n* = 111/174, 64%) affiliated with an institution in Europe (*n* = 77/174, 45%), comprised non-Cochrane reviews (*n* = 165/174, 95%) published in the last two years (*n* = 85/174, 49%) and were registered in PROSPERO (*n* = 125/174, 72%) (Table 2). These SRs included a median of 11 studies (interquartile range (IQR): 7 to 17) and had a low precision of searches (median: 2%, IQR: 1 to 4%). There were no explicit language restrictions (*n* = 135/174, 80%) nor explicit exclusion of non-English studies (*n* = 170/174, 98%) in most SRs. Only 37 (21%) SRs involved a librarian or/and search specialist during the conduct. More than half of the SRs performed a meta-analysis (*n* = 97/174, 56%). The systematic reviews were mostly interventional (*n* = 146/174, 84%) and pertained to human participants (*n* = 156/174, 91%). Overall, the characteristics were similarly distributed between SRs with and without non-English studies. However, SRs with non-English studies had slightly more studies (median: 13, IQR: 9 to 22) than SRs without such studies (median: 10, IQR: 7 to 17).

### Association of including non-English studies with several characteristics

The multivariable binary logistic regression revealed an association of substantial magnitude between the inclusion of non-English studies and various characteristics of the SRs (Table 2). However, the association was not statistically significant for all characteristics apart from the restriction on language: the odds of including a non-English study were lower by 89% (OR: 0.11, 95% CI: 0.01 to 0.55, P = 0.004) in SRs posing a language restriction. Though statistically non-significant, SRs including fewer than four authors, originating from America, being non-Cochrane, having a protocol registration, published in earlier years, involving a librarian, conducting a meta-analysis and specialising in interventional studies were associated with higher odds of including non-English studies (OR range: 1.13 to 3.66) than SRs with more authors, originating from other continents, published recently, being Cochrane SRs, without a protocol registration, a librarian, or a meta-analysis and specialising in other study types.

PRISMA 2020 flow diagram for new systematic reviews which included searches of databases and registers only

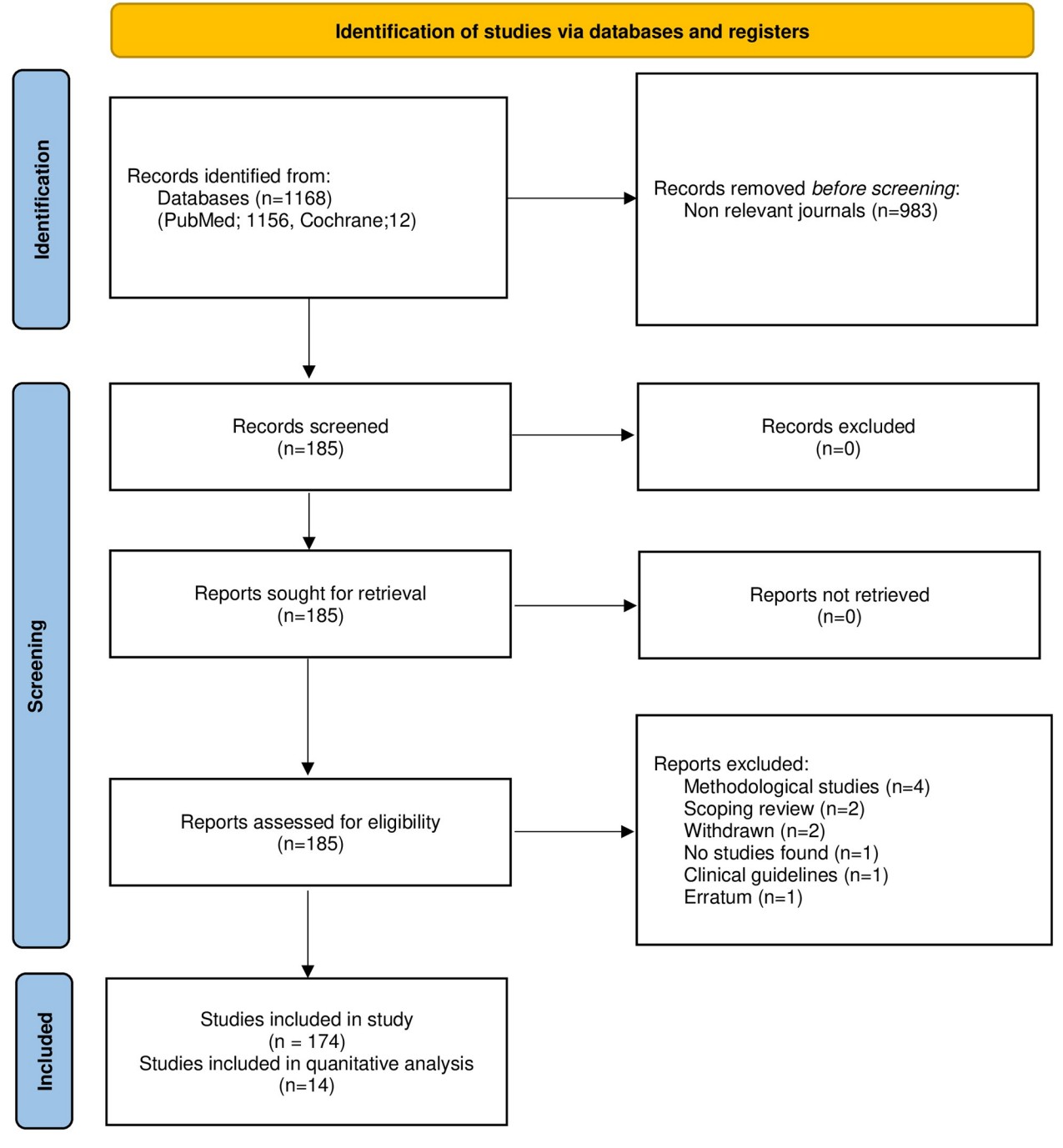

**Fig 1. PRISMA flow diagram for the included systematic reviews.**

**Table 2. Characteristics of systematic reviews with and without studies on non-English languages.**

| Characteristic | Non-English language included | | | OR (95% CI)* | p-value |
|---|---|---|---|---|---|
| | Overall (n = 174) | No (n = 129) | Yes (n = 45) | | |
| Authors Number | | | | | |
| 1–3 | 43 (25%) | 30 (23%) | 13 (29%) | reference | |
| 4–6 | 111 (64%) | 84 (65%) | 27 (60%) | 0.71 (0.27, 1.93) | 0.50 |
| 7–9 | 20 (11%) | 15 (12%) | 5 (11%) | 0.64 (0.14, 2.64) | 0.54 |
| Continent | | | | | |
| America | 46 (26%) | 33 (26%) | 13 (29%) | reference | |
| Asia & others | 51 (29%) | 40 (31%) | 11 (24%) | 0.67 (0.21, 2.04) | 0.48 |
| Europe | 77 (45%) | 56 (43%) | 21 (47%) | 0.91 (0.35, 2.37) | 0.85 |
| Review type | | | | | |
| Cochrane | 9 (5%) | 6 (5%) | 3 (7%) | reference | |
| non-Cochrane | 165 (95%) | 123 (95%) | 42 (93%) | 3.66 (0.60, 25.81) | 0.16 |
| Publication Year | | | | | |
| 2017 | 28 (16%) | 17 (13%) | 11 (24%) | 0.81 (0.60, 1.08) | 0.15 |
| 2018 | 32 (18%) | 26 (20%) | 6 (13%) | | |
| 2019 | 29 (17%) | 22 (17%) | 7 (16%) | | |
| 2020 | 37 (21%) | 25 (19%) | 12 (27%) | | |
| 2021 | 48 (28%) | 39 (30%) | 9 (20%) | | |
| PROSPERO registration | | | | | |
| No | 49 (28%) | 42 (33%) | 7 (16%) | reference | |
| Yes | 125 (72%) | 87 (67%) | 38 (84%) | 1.64 (0.58, 5.00) | 0.36 |
| Number of included studies | | | | | |
| Median (IQR) [range] | 11.00 (7.00, 17.00) [2.00, 94.00] | 10.00 (7.00, 16.50) [2.00, 94.00] | 13.00 (9.00, 22.00) [4.00, 63.00] | 1.02 (0.99, 1.05) | 0.19 |
| The precision of the search[1] | | | | | |
| Median (IQR) [range] | 0.02 (0.01, 0.04) [0.001, 0.27] | 0.02 (0.01, 0.04) [0.001, 0.27] | 0.02 (0.01, 0.07) [0.001, 0.26] | 1.04 (0.77, 1.40) | 0.82 |
| Number of included non-English languages[1] | | | | | |
| Median (IQR) [range] | 1 (1, 3) [1, 19] | NA | 1 (1, 3) [1, 19] | NA | NA |
| Restriction on language[1] | | | | | |
| No | 135 (80%) | 91 (74%) | 44 (98%) | reference | |
| Yes | 33 (20%) | 32 (26%) | 1 (2%) | 0.11 (0.01, 0.55) | 0.004 |
| Excluded studies on non-English language | | | | | |
| No | 170 (98%) | 125 (97%) | 45 (100%) | reference | |
| Yes | 4 (2%) | 4 (3%) | 0 (0%) | 0.03 (0.01, 1.47) | 0.08 |
| Librarian/search specialist | | | | | |
| No | 137 (79%) | 103 (80%) | 34 (76%) | reference | |
| Yes | 37 (21%) | 26 (20%) | 11 (24%) | 2.98 (0.97, 9.46) | 0.06 |
| Meta-analysis performed | | | | | |
| No | 77 (44%) | 63 (49%) | 14 (31%) | reference | |
| Yes | 97 (56%) | 66 (51%) | 31 (69%) | 1.92 (0.79, 4.91) | 0.15 |
| Type of systematic review | | | | | |
| Diagnostic | 9 (5%) | 8 (6%) | 1 (2%) | reference | |
| Epidemiological | 19 (11%) | 18 (14%) | 1 (2%) | 0.48 (0.03, 7.50) | 0.58 |
| Interventional | 146 (84%) | 103 (80%) | 43 (96%) | 2.32 (0.38, 25.68) | 0.38 |

(*Continued*)

**Table 2.** (Continued)

| Characteristic | Non-English language included | | | OR (95% CI)* | p-value |
|---|---|---|---|---|---|
| | Overall (n = 174) | No (n = 129) | Yes (n = 45) | | |
| Type of included studies | | | | | |
| Animal | 12 (7%) | 7 (5%) | 5 (11%) | reference | |
| Human | 159 (91%) | 119 (92%) | 40 (89%) | 0.29 (0.05, 1.45) | 0.13 |
| In vitro | 3 (2%) | 3 (3%) | 0 (0%) | 0.08 (0.01, 1.74) | 0.11 |

CI, confidence interval; IQR, interquartile range; NA, not applicable; OR, odds ratio

*Strong evidence when the 95% confidence interval excludes 1 (p-value < 5%); otherwise, weak evidence for the corresponding association.

[1] 'Precision of the search' and 'Number of included non-English languages' were not reported in one systematic review. 'Restriction on language' was not reported in six systematic reviews.

### Frequency of non-English languages

Fig 2 presents the frequency of non-English studies based on non-English language and SR type. The median number of non-English studies was one with an IQR of 1 to 3. The most commonly used language was Chinese, followed by Portuguese and German. Non-Cochrane SRs included more non-English studies than Cochrane SRs. One Cochrane SR included several non-English languages (Farsi, French, Italian, and Portuguese). Five Cochrane SRs considered two non-English languages (Chinese and Portuguese, Chinese and Turkish, German and Portuguese, or Dutch and German), and one Cochrane SR included three non-English studies (Chinese, Portuguese, and Turkish).

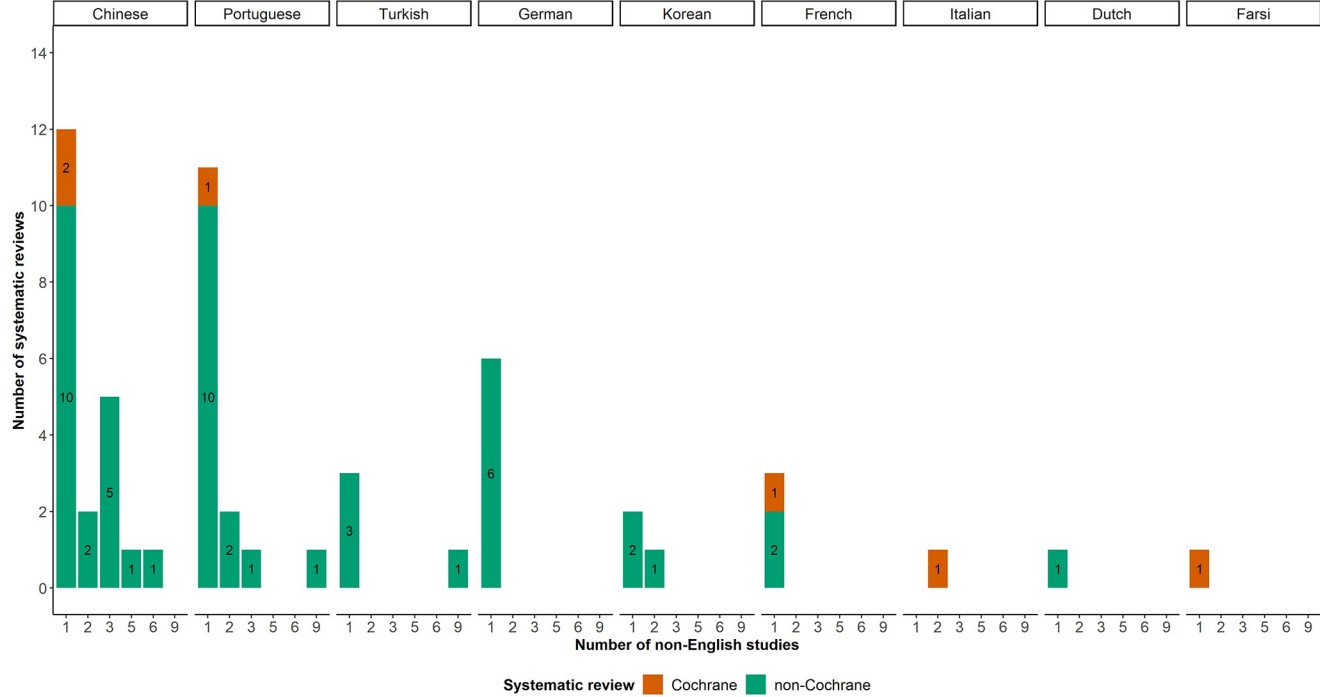

**Fig 2. Bar plots with the number of non-English studies for each non-English language and systematic review type.**

**Table 3. The association of statistical significance with the inclusion of non-English studies.**

| Non-English language included | Statistical significance | | Total | OR, (95%CI) | P value |
|---|---|---|---|---|---|
| | Yes | No | | | |
| **No** | 5 | 4 | 9 | reference | |
| **Yes** | 16 | 4 | 20 | 3.20 (0.57, 18.92) | 0.18 |

### Statistical significance and inclusion of non-English studies

Of the 29 SRs that conducted a meta-analysis and reported the language of the synthesised studies, 69% ($n$ = 20) included at least one non-English study; of those meta-analyses, 80% (16/20) provided a statistically significant effect. (Table 3) The univariate binary logistic regression indicated that the odds of statistical significance in the summary effect estimate was 3.20 times larger in systematic reviews with non-English studies than in systematic reviews with only English studies. However, the association was statistically non-significant (OR:3.20, 95% CI: 0.57, 18.92, P = 0.18).

### Examining the influence of non-English studies on summary results

Out of the sample, only fourteen meta-analyses comprised the dataset of the meta-epidemiological study. The supplementary material provides descriptive statistics on the number of studies and sample size of the included meta-analyses. The meta-epidemiological analysis revealed that non-English studies tended to overestimate the SMD by approximately 0.30 on average compared to English studies (Fig 3). Specifically, the difference of SMD was not statistically significant ($\Delta$SMD = -0.29, 95%CI: -0.63 to 0.05, P = 0.37) using a random-effects model due to substantial statistical heterogeneity, but it was spuriously statistically significant with a fixed-effect model ($\Delta$SMD = -0.31, 95%CI: -0.58 to -0.05, P = 0.03) (Fig 3). The $\Delta$SMD was positive in four meta-analyses, implying a larger SMD from English studies. Overall, including non-English studies improved the precision of the summary SMD, whilst including only English studies decreased SMD substantially in some meta-analyses (S1 Fig). Most meta-analyses were associated with fairly high or extreme statistical heterogeneity, regardless of language restriction (S2 Fig). Restricting inclusion to English studies increased statistical heterogeneity in almost half meta-analyses compared to including all studies regardless of language (S2 Fig).

## Discussion

### Evidence summary

In general, 26% of orthodontic SRs included non-English language studies in this assessment, which may lead to language bias. There was evidence of language bias also in other medical specialties [23], though the proportion of included non-English studies was lower compared to the orthodontics field (7.4% versus 26%) [24]. This could be related to the fact that less than half of the non-English medical studies are indexed in Medline, and 68% of orthodontic SRs searched Medline [2]. The most frequently included non-English language in orthodontic SRs was the Chinese language, which is in disagreement with the findings of Cochrane reviews as the Chinese studies were the least included in two medical domains [23]. However, the Chinese language comprised 47% of the medical literature published in languages other than English, followed by German and Spanish studies [24].

The language restriction in the search decreased the likelihood of including non-English studies in SRs by 89%, which is a reasonable finding. However, the majority of SRs performing

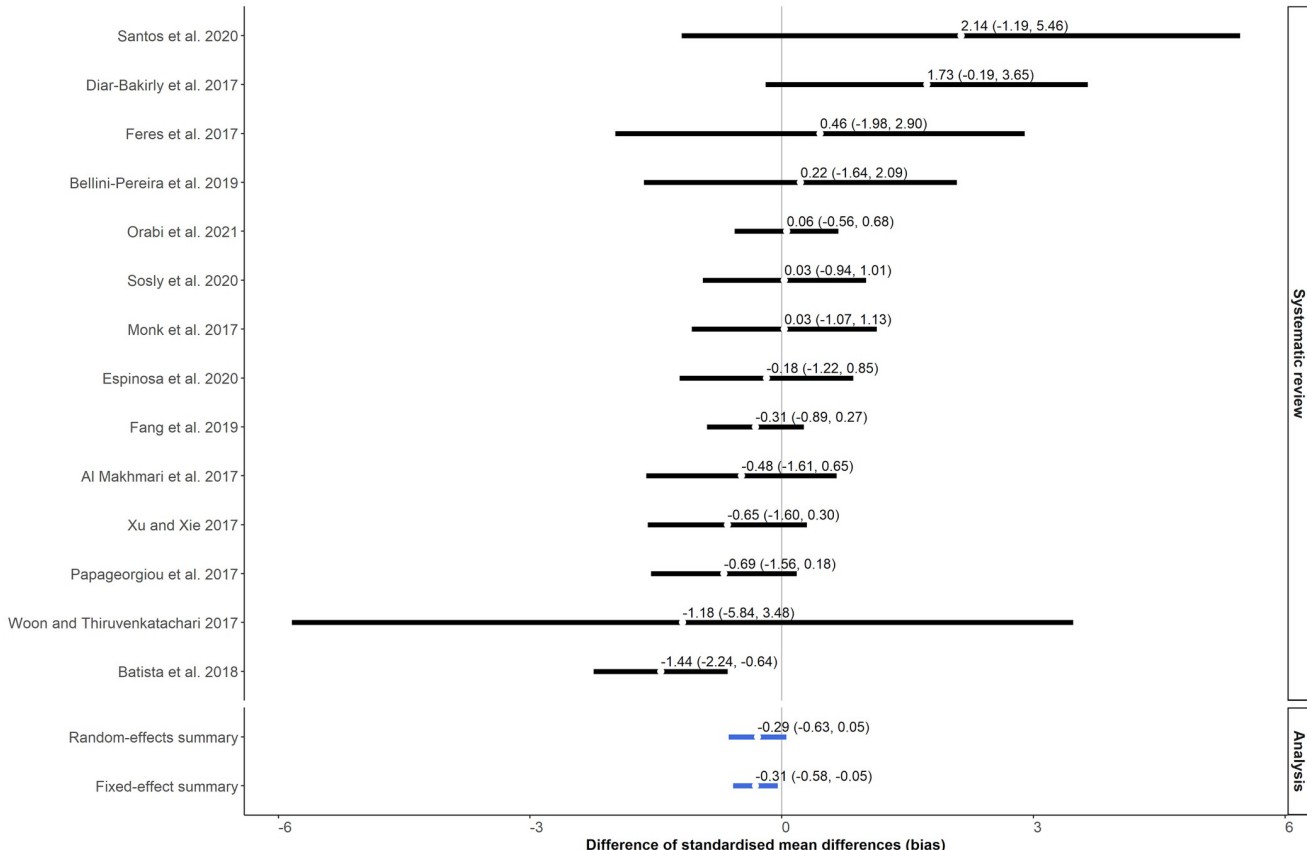

**Fig 3. Forest plot showing the difference in standardised mean difference (ΔSMD) between non-English and English studies using a set of meta-analysis.** *Negative bias favours the English studies.

no language restriction did not include a non-English study (67%; 91/135) and four SRs excluded non-English studies after the search process. That could be attributed to limiting the eligibility criteria to English studies without limiting the search strategy, rendering the conduct of SR unnecessarily time-consuming and requiring additional human and financial resources. As such, if only English studies were included in the SR, restricting the search to only English studies would be a valid option and saving factor for time, human and financial recourses. In this approach, non-English records would be removed from the search results, thus, reducing the number of retrieved reports and, subsequently, the initial screening time and efforts. In this regard, manual search, particularly in google scholar may result in non-English studies even if the search was restricted in the other bibliographies. This was evident in one included SR [25] with additional manual search, although the search was restricted in the searching bibliographies.

Furthermore, language bias in SRs may not result only from language restrictions in search strategy or eligibility criteria. It could be due to the information sources used in the search. For instance, MEDLINE, Embase, and CINAHL are international English bibliographic databases which comprise only limited non-English literature. In contrast, special databases can be used for non-English studies [26].

The present study found a statistically non-significant association between including non-English studies in orthodontic SRs and SRs with small co-authorship or SRs led by an author

affiliated in American institute. International collaborations including authors speaking several languages may be a possible explanation for this finding. Text screening, risk of bias assessment, and data extraction require an expert researcher who understands the language of the retrieved articles. In this respect, Cochrane can hire translators who are not necessarily directly involved in the reviews (https://www.cochrane.org/join-cochrane/translate).

It is well established that SRs with a registered protocol have a higher quality than non-registered SRs [27]. Our study revealed that SRs with a protocol registration were more likely to include non-English studies. The inclusion of non-English studies also coincided with a higher quality of SRs when performing the unrestricted comprehensive search as provided by the AMSTAR2 tool [28].

We found that meta-analyses with non-English studies were more likely to yield statistical significance than meta-analyses that included only English studies. However, this association was not statistically significant due to the small analysed sample, which may have led to insufficient power to detect the difference. Relevant empirical studies on several medical fields uncovered that excluding non-English studies from SRs impacted the meta-analysis results, with the change in the effect estimate varying from negligible to large [15, 23]. Moreover, the statistical heterogeneity was reduced after removing non-English studies from the meta-analyses [29, 30]. Our meta-epidemiological analysis demonstrated a larger summary effect size from including non-English studies, though this was not statistically significant due to material statistical heterogeneity. Study characteristics such as risk of bias and sample size may have contributed to the heterogeneity in ΔSMDs observed across the meta-analyses. Investigating sources of heterogeneity that may explain the difference in treatment effects between non-English and English studies was out of this study scope. The analysis also found that the statistical heterogeneity was reduced in almost half of meta-analyses when both non-English and English studies were included (S2 Fig).

Moreover, different studies [24, 31–33] assessed the quality of the English and non-English studies, and found a higher risk of bias in non-English studies due to suboptimal randomisation, insufficient reporting of the blinding, and incomplete data. As such, non-English studies may be removed from the meta-analysis in the context of a sensitivity analysis to inspect the robustness of the meta-analysis results.

Though including non-English studies in SR may be laborious in terms of time and resources [15], it is still an imperative option in the case of trials' paucity. For instance, one SRs [34] in our sample included only non-English studies. Likewise, including non-English studies has a crucial impact when the disease of interest has a different prevalence between ethnicities. For example, class III malocclusion is more prevalent in Asian people than in other ethnicities [35], rendering studies from these regions important in providing valuable information. On a positive note, authors' collaboration may facilitate including these studies in the SRs if at least one expert author in the language is involved.

## Limitations and strength

The search in this study was restricted to two databases and SRs published in the last five years, possibly missing some relevant studies. A wider search may have some impact on the importance of including non-English studies. For instance, it may turn statistically non-significant results into statistically significant, particularly when more meta-analyses with non-English language are included in the meta-epidemiological study. Besides, this study attempted to map the problem and merely explore the implications of language bias on the meta-analysis results. The bias associated with including non-English studies may be confounded by the quality of the conduct, analysis and reporting of these studies. We did not investigate the quality of the

SRs as a function of language bias since it was out of the scope. Three experienced investigators implemented the study selection and the data acquisition, and the analysis was conducted by an experienced biostatistician using state-of-the-art methods, ensuring a high-quality meta-epidemiological study. The authors were concerned about the unaddressed correlation that may arise from including more than one meta-analysis from the same SR in the meta epidemiological analysis. As such, only one meta-analysis with non-English study was involved. The prior protocol was not registered due to the constraints imposed by the COVID period, which may introduce some bias. However, the authors diligently adhered to their protocol and rigorous guidelines.

## Conclusion

Language restriction seems to have no statistically significant impact on the results of orthodontic SRs. Including non-English studies in orthodontic SRs aligns with the scope of systematic review to retrieve all relevant evidence and may also increase the statistical power due to the increase in the sample of studies. Furthermore, language inclusiveness may aid in gauging the quality of the evidence base and uncovering knowledge gaps to prioritise future research. For instance, research questions addressed exclusively by non-English studies of questionable quality render the evidence at hand restrictive and inappropriate for guideline recommendations. Then, a research agenda may be set up to answer the research question anew using a living systematic review that is constantly updated with new studies and poses no language restriction. However, authors should bear in mind that non-English studies may have a higher risk of bias and should assess the appropriateness of non-English studies individually according to the specific topic of the review.

## Supporting information

**S1 Table. Studies excluded with the reason.**
(DOCX)

**S1 File. Technical notes.**
(DOCX)

**S1 Fig. Forest plots on the summary standardised mean difference and 95% confidence interval in 14 meta-analyses.**
(DOCX)

**S2 Fig. Line plot on the between-study standard deviation (x-axis) in 14 meta-analyses (y-axis).**
(DOCX)

## Author Contributions

**Conceptualization:** Samer Mheissen.

**Data curation:** Samer Mheissen, Baraa Daraqel, Ahmad Saleem Alsafadi.

**Formal analysis:** Samer Mheissen, Loukia M. Spineli.

**Investigation:** Loukia M. Spineli, Baraa Daraqel, Ahmad Saleem Alsafadi.

**Methodology:** Samer Mheissen, Loukia M. Spineli, Baraa Daraqel.

**Resources:** Baraa Daraqel, Ahmad Saleem Alsafadi.

**Software:** Loukia M. Spineli.

**Supervision:** Loukia M. Spineli.

**Visualization:** Loukia M. Spineli.

**Writing – original draft:** Samer Mheissen, Loukia M. Spineli, Ahmad Saleem Alsafadi.

**Writing – review & editing:** Samer Mheissen, Loukia M. Spineli, Baraa Daraqel, Ahmad Saleem Alsafadi.

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
