## [Decision Letter · Decision Letter 0]

21 Nov 2023

PONE-D-23-28166Language Bias in Orthodontic Systematic Reviews: A Meta-epidemiological StudyPLOS ONE

Dear Dr. Mheissen,

Thank you for submitting your manuscript to PLOS ONE. After careful consideration, we feel that it has merit but does not fully meet PLOS ONE’s publication criteria as it currently stands. Therefore, we invite you to submit a revised version of the manuscript that addresses the points raised during the review process.

We look forward to receiving your revised manuscript.

Kind regards,

Luiz Alexandre Chisini, Ph.D

Academic Editor

PLOS ONE

Journal Requirements:

1. When submitting your revision, we need you to address these additional requirements. Please ensure that your manuscript meets PLOS ONE's style requirements, including those for file naming. The PLOS ONE style templates can be found at https://journals.plos.org/plosone/s/file?id=wjVg/PLOSOne_formatting_sample_main_body.pdf and https://journals.plos.org/plosone/s/file?id=ba62/PLOSOne_formatting_sample_title_authors_affiliations.pdf 2. We note that you have referenced (Hartling L, Featherstone R, Nuspl M, Shave K, Dryden DM, Vandermeer B. Grey) which has currently not yet been accepted for publication. Please remove this from your References and amend this to state in the body of your manuscript: (Hartling L, Featherstone R, Nuspl M, Shave K, Dryden DM, Vandermeer B. Grey [Unpublished]”) as detailed online in our guide for authorshttp://journals.plos.org/plosone/s/submission-guidelines#loc-reference-style. 3. Please include captions for your Supporting Information files at the end of your manuscript, and update any in-text citations to match accordingly. Please see our Supporting Information guidelines for more information: http://journals.plos.org/plosone/s/supporting-information. 

Additional Editor Comments:

The article's proposal is interesting and innovative; however, the authors need to clarify some points of the methods and need to revise the interpretations, that are not accurate.

Abstract:

- The sentence should be revised: “The association between SRs characteristics and including non-English studies was overall statistically inconclusive apart from the restriction on language: the odds of including a non-English study were reduced by 89% in SRs with a language”. Was the result associated or not? please present the OR and 95% CI.

- In the sentence: “The evidence was inconclusive due to substantial statistical heterogeneity using a random-effects model, but it was spuriously conclusive with a fixed-effect model”. Inconclusive is not the best way to interpretate the p-value. Please revise all the paper and the interpretations of this result. Moreover, fixed-model is only a sensitivity analysis and should be only discussed in the discussion.

- Authors report that “174 SRs were eligible for inclusion in this study.”. However, this can lead the readers to wrong interpretations since the authors had two different analyses: i) SR and ii) meta-analysis. So, only 14 studies were included in the meta-analysis. This should be clear in the abstract.

Methods:

- Why did de authors not register a protocol? This introduces an important bias in the study

- Authors report “[…] in orthodontic journals with the highest impact factor (2021):”. Was there a cut-off? The authors need to clarify the criteria for each covariable used in the study.

- Why only two databases were searched?

- Considering that the statistical tests used by the authors are based on the assumption of independence of observations, did the authors investigate whether the studies included in each meta-analysis are from unique studies? Authors need to ensure that the independence of observations is not violated. So, if one meta-analysis study was included in more than one study, the results lose independence.

- Authors reported that: “One primary outcome was selected from each SR that included at least one non-English study to investigate the impact of the non-English language on the meta-analysis results.”; what were the criteria for standardizing this selection?

- Authors mention that did a meta-regression ”Initially, a random-effects meta-regression was

conducted to estimate the difference in standardised”, However, I didn’t find this in the paper. Please, revise it.

Results:

- It is recommended to cite the references of excluded studies in the sentence: “Of the 11 excluded SRs, four comprised methodological studies, two were scoping reviews, two were withdrawn, one did not find any eligible studies, one was a clinical guideline, and one was an erratum.”

- Please, highlight that only 14 studies were included as meta-analyses to avoid misinterpretation by the readers.

- Revise the paper and don’t use “statistically inconclusive“ as in the sentence: “the association was statistically inconclusive (p-value > 0.05)”. It is not inconclusive. In fact, there is no association.

- Is from which analysis the results “the odds of including a non-English study were reduced by 89% (95% CI: 45% to 99%)”? Because I didn’t find these results in the tables.

- In the sentence “80% provided a conclusive summary effect estimate (p-value < 0.05)”, please, change “conclusive” to “association”

- Did the authors include 14 or 20 studies in the meta-analysis evaluation? Because sometimes the authors' reporting is 20 and sometimes 14 In the text.

- In the sentence “Most meta-analyses were associated with fairly high or extreme statistical heterogeneity” how did the authors estimate the heterogeneity of the studies? This is not mentioned in the results. Please, consider that I2 cannot estimate the heterogeneity, but I2 represents the percentage of observed variability that is attributed to the variability of true effects.

Discussion

- Almost all results and discussions are misinterpreted. For example: The sentence “The present study found positive association between including non-English studies in orthodontic SRs and SRs with small co-authorship or SRs led by an author affiliated in American institute, but the evidence was inconclusive.” is not accurate. There was no association. Please, revise all the paper and correct it. Also here: “We found that meta-analyses with non-English studies were more likely to yield statistical significance than meta-analyses that included only English studies. However, the evidence was inconclusive due to the small analysed sample”. There was no association. Moreover, for meta-analysis results the authors should present power calculation and discuss it. Probably the results were not associated due to a lack of power and this is related to the limitations that authors impose on papers: 2017 to 2021. So, this results in a limited number of meta-analyses with at least one non-English study. Considering that the authors had two aims, they could increase the period for studies to be included in the meta-analysis to provide an “n” of studies adequate.

- The limitations of the study must be deeply discussed.

Figure 3 has a low quality. Please add a high-quality size in Figure 3.

Reviewers' comments:

Reviewer's Responses to Questions

**Comments to the Author**

1. Is the manuscript technically sound, and do the data support the conclusions?

Reviewer #1: Yes

Reviewer #2: Partly

2. Has the statistical analysis been performed appropriately and rigorously? 

Reviewer #1: Yes

Reviewer #2: I Don't Know

3. Have the authors made all data underlying the findings in their manuscript fully available?

Reviewer #1: No

Reviewer #2: Yes

4. Is the manuscript presented in an intelligible fashion and written in standard English?

Reviewer #1: Yes

Reviewer #2: Yes

5. Review Comments to the Author

Reviewer #1: In general terms, I found the article's proposal interesting, innovative, and well-executed by the authors. Language Bias should be a source of concern within evidence synthesis, given that other studies have already shown how geographical differences can impact submission and publication processes mainly in high-impact journals. However, I believe that some further clarification is needed.

Introduction:

Paragraphs 3 and 4 (page 3) present studies that have addressed Language Bias in some way but the definition of Language Bias is unclear in the text. I suggest including a brief definition of Language Bias and the context of English as the dominant scientific spoken/written language in Dentistry.

Methods:

In the Data Collection Process subsection (page 4): Were there agreement measurements between two reviewers in the pilot assessment of SR?

Just another small note on the description of the variables collected (page 4 - Data Collection Process subsection): How was the continent variable collected and categorized? Did you consider the affiliation of the first author?

In the Statistical Analysis and Data Synthesis subsection (page 5): In the multivariable binary logistic regression, were all the variables of interest included in the final multivariable regression model? Was there any specific way of selecting variables for the final model? Also, the level of significance (p-value >0.05) could be added here instead in the results section.

Results:

If possible, present the list of the included systematic reviews as supplementary material to accomplish with the data availability statement.

Discussion:

In the sentence: “The present study found positive association between including non-English studies in orthodontic SRs and SRs with small co-authorship or SRs led by an author affiliated with an American institute, but the evidence was inconclusive.” (Page 10, 4th paragraph): I believe that an improvement could be made to the wording, considering that "positive association" could be interpreted as a p<0.05, which was not the case in the study. I suggest changing the wording to "a relationship between ..." or some similar term at the authors' discretion.

Regarding limitations and strength subsection: In the metanalysis, could the effect measures found be affected by the heterogeneity of outcomes considered as the main outcome? Could this be considered a limitation of this study?

Reviewer #2: Dear Editor Luiz Alexandre Chisini,

Thank you for the opportunity to review this article.

I hope you find my feedback valuable.

Major Review

This article aims to assess language bias in systematic reviews published in orthodontics;

however, it may incorporate other biases at various levels:

Searches were conducted only in two databases. While I understand that the majority

of relevant literature might be present in these databases, this choice needs a more

robust justification. A sample calculation can be used to justify the number of articles that need to be included, potentially reducing the need for searches in additional databases.

There is also a significant date restriction for the search that needs clarification

The most critical point is that there has been no prior publication of the study protocol.

Minnor Review

Other minor points that also require attention:

It would be beneficial to make it clearer what the primary outcome of this study is.

From what I understand, the analysis considered studies that included both non-English language articles and those that did not. In the results section, the authors report that there were no explicit language restrictions ( = 135, 80%) or explicit exclusion of non-English studies ( = 170, 98%) in the majority of systematic reviews. However, there is a possibility that some systematic reviews may include studies in other languages, even if these studies do not exist and, consequently, are not found. How would they be considered in the analysis? Perhaps, this potential scenario could be addressed in the discussion.

In the results section, when indicating the p-value, please provide the exact number rather than using p-value > 0.05. If the p-value is very small, describe it as p-value < 0.0001.

I also suggest including the "numerical" values in the results presented in the abstract.

Please define what would constitute an inconclusive association. Was there no

association? What is the power of the test used to prevent Type 1 error?

In the discussion, the publication bias can be explored. As the authors themselves point

out, studies in other languages may exhibit lower methodological quality.

Additionally, it is essential to provide the search strategy used for the Cochrane Library

database.

The theme of this study is very interesting. With some corrections to strengthen the methodological robustness, there is a good potential for publication.

6. PLOS authors have the option to publish the peer review history of their article (what does this mean?). If published, this will include your full peer review and any attached files.

Reviewer #1: No

Reviewer #2: **Yes: **Jaqueline Barbieri Machado

---

## [Author Response · Author response to Decision Letter 0]

5 Dec 2023

Dear editor Luiz Alexandre Chisini, 

We would like to thank you and the reviewers for the comments and suggestions that we think will improve our manuscript. 

Please find below responses and actions taken. In the revised manuscript we highlight amended sections.

PONE-D-23-28166

Language Bias in Orthodontic Systematic Reviews: A Meta-epidemiological Study

PLOS ONE

We look forward to receiving your revised manuscript.

Kind regards,

Luiz Alexandre Chisini, Ph.D

Academic Editor

PLOS ONE

Journal Requirements:

Authors’ response 

We have amended the manuscript accordingly.

2. We note that you have referenced (Hartling L, Featherstone R, Nuspl M, Shave K, Dryden DM, Vandermeer B. Grey) which has currently not yet been accepted for publication. Please remove this from your References and amend this to state in the body of your manuscript: (Hartling L, Featherstone R, Nuspl M, Shave K, Dryden DM, Vandermeer B. Grey [Unpublished]”) as detailed online in our guide for authors

http://journals.plos.org/plosone/s/submission-guidelines#loc-reference-style.

Authors’ response 

We have checked the reference and it indicates a published study in BMC Medical research Methodology Journal in 2017 (https://bmcmedresmethodol.biomedcentral.com/articles/10.1186/s12874-017-0347-z). if the editor think that we should refer it as unpublished we will do that.

Authors’ response 

We have amended the manuscript accordingly.

Additional Editor Comments:

The article's proposal is interesting and innovative; however, the authors need to clarify some points of the methods and need to revise the interpretations, that are not accurate.

Abstract:

- The sentence should be revised: “The association between SRs characteristics and including non-English studies was overall statistically inconclusive apart from the restriction on language: the odds of including a non-English study were reduced by 89% in SRs with a language”. Was the result associated or not? please present the OR and 95% CI.

Authors’ response 

There was no statistically significant association except for the language restriction. The abstract has been amended accordingly:

‘The association between SRs characteristics and including non-English studies was not statistically significant except for the restriction on language: the odds of including non-English studies reduced by 89% in SRs with a language restriction (OR: 0.11, 95%CI: 0.01 0.55, P<0.004)’

- In the sentence: “The evidence was inconclusive due to substantial statistical heterogeneity using a random-effects model, but it was spuriously conclusive with a fixed-effect model”. Inconclusive is not the best way to interpretate the p-value. be only discussed in the discussion. Please revise all the paper and the interpretations of this result. Moreover, fixed-model is only a sensitivity analysis and should be only discussed in the discussion.

Authors’ response 

The fixed effect model was removed from abstract, and the abstract has been amended accordingly:

‘As such, the evidence of larger meta-analysis results with non-English studies is inconclusive.’

- Authors report that “174 SRs were eligible for inclusion in this study.”. However, this can lead the readers to wrong interpretations since the authors had two different analyses: i) SR and ii) meta-analysis. So, only 14 studies were included in the meta-analysis. This should be clear in the abstract.

Authors’ response 

We have added the following clarity to the results:

‘Out of the sample, only fourteen meta-analyses were included in the meta-epidemiological analysis’ 

Methods:

- Why did de authors not register a protocol? This introduces an important bias in the study

Authors’ response 

There was a prior protocol to ensure a well-conducted study. Unfortunately, the authors did not register it on PROSPERO due to the constraints imposed by the COVID period and the simultaneous need for prompt initiation of the study. We have included this point in the study limitations:

‘The prior protocol was not registered due to the constraints imposed by the COVID period, which may introduce some bias. However, the authors diligently adhered to their protocol and rigorous guidelines.’

- Authors report “[…] in orthodontic journals with the highest impact factor (2021):”. Was there a cut-off? The authors need to clarify the criteria for each covariable used in the study.

Authors’ response 

The cut-off was the five leading orthodontic journals. We have added the impact factor of these journals to the methods:

‘American Journal of Orthodontic and Dentofacial Orthopedics (3.6), European Journal of Orthodontics (4.3), Progress in Orthodontics (3.7), Angle Orthodontist (3.4), and Orthodontics & Craniofacial Research (2.8).’ 

- Why only two databases were searched?

Authors’ response

This was discussed between the authors before conducting the search due to its importance. Our understanding was to include the largest number of published systematic reviews in these journals for the last five years. The search in PubMed was undertaken to retrieve systematic reviews published in the leading orthodontic journals which is indexed in PubMed. While Oral health group was searched to retrieve the relevant Cochrane reviews.

We added the last two sentences of the above justification to the Methods.

- Considering that the statistical tests used by the authors are based on the assumption of independence of observations, did the authors investigate whether the studies included in each meta-analysis are from unique studies? Authors need to ensure that the independence of observations is not violated. So, if one meta-analysis study was included in more than one study, the results lose independence.

Authors’ response

The authors ensured the independence of the data between the included meta-analyses by including unique data in each meta-analysis. If there was a violation of data structure by including correlated data in the same meta-analysis, the authors calculated the correlation and adjusted the standard error according to the Cochrane handbook; Chapter 23. Details were provided in the supplementary materials.

- Authors reported that: “One primary outcome was selected from each SR that included at least one non-English study to investigate the impact of the non-English language on the meta-analysis results.”; what were the criteria for standardizing this selection?

Authors’ response

This paragraph was amended according to the editor's comment as follows:

‘One outcome was selected from each SR that included at least one non-English study to investigate the impact of the non-English language on the meta-analysis results. Specifically, we considered the following ‘algorithm’: if the meta-analysis of primary outcome included at least three studies with at least one being non-English, this meta-analysis was included in our collection. If more meta-analyses were eligible, we opted for the first one addressing the primary outcome. If the meta-analysis of the primary outcome was not eligible, the meta-analysis for secondary outcomes was checked.’

- Authors mention that did a meta-regression ”Initially, a random-effects meta-regression was

conducted to estimate the difference in standardised”, However, I didn’t find this in the paper. Please, revise it.

Authors’ response

This was explained in the supplementary materials due to the limited word counts as follows:

‘A two-stage approach was performed to analyse the meta-epidemiological data and estimate the average bias attributed to non-English studies. The meta-epidemiological data comprised meta-analyses on the primary outcome (one from each eligible SR) that included at least three studies, with at least one being non-English. The standardised mean difference (SMD) was the effect measure since all selected meta-analyses referred to a continuous primary outcome. Initially, a random-effects meta-regression with language type (non-English versus English) as the covariate was conducted in each meta-analysis. The intercept referred to the summary SMD in English studies, and the slope measured the summary difference in SMD (ΔSMD) between non-English and English studies. This model allowed the between-study variance to be estimated separately for each subgroup with at least two studies using the restricted maximum likelihood estimator for heterogeneity. Then, a random-effects meta-analysis was performed to combine the ΔSMDs across the meta-analyses. A negative ΔSMD would indicate that non-English studies overestimated SMD.’

Results:

- It is recommended to cite the references of excluded studies in the sentence: “Of the 11 excluded SRs, four comprised methodological studies, two were scoping reviews, two were withdrawn, one did not find any eligible studies, one was a clinical guideline, and one was an erratum.”

Authors’ response

We have found that citing erratum and withdrawn articles is difficult, so we have added a supplementary table with the details of these articles.

- Please, highlight that only 14 studies were included as meta-analyses to avoid misinterpretation by the readers.

Authors’ response

The text was amended according to the editor’s comment:

‘Out of the sample, only fourteen meta-analyses comprised the dataset of the meta-epidemiological study.’

- Revise the paper and don’t use “statistically inconclusive“ as in the sentence: “the association was statistically inconclusive (p-value > 0.05)”. It is not inconclusive. In fact, there is no association.

Authors’ response

We have revised these statements throughout the paper and all variables related to statistical significance were amended. We kept only related words to the discussion. In summary, when p-value > 0.05; we concluded the association to be statistically non-significant; otherwise, the association was statistically significance. 

- Is from which analysis the results “the odds of including a non-English study were reduced by 89% (95% CI: 45% to 99%)”? Because I didn’t find these results in the tables.

Authors’ response

We are sorry for this confusion. The value 89% came from subtracting an OR of 0.11 from 1, and the same approach was used to obtain the percentages in the bounds of the 95% CI of the OR. We have added these values between brackets in the proper place:

‘the odds of including a non-English study were lower by 89% (OR: 0.11, 95% CI: 0.01 0.55, P<0.004) in SRs posing a language restriction.’

- In the sentence “80% provided a conclusive summary effect estimate (p-value < 0.05)”, please, change “conclusive” to “association”

Authors’ response

This was amended accordingly.

- Did the authors include 14 or 20 studies in the meta-analysis evaluation? Because sometimes the authors' reporting is 20 and sometimes 14 In the text.

Authors’ response

20 systematic reviews conducted meta-analysis and included at least one non-English study. However, only 14 SRs were included in the meta-epidemiological analysis.

 we have added table3 to clarify the association between the statistical significance and the inclusion of non-English studies. Also, we have added the following text under ‘Examining the influence of non-English studies on summary results’:

‘Out of the sample, only fourteen meta-analyses comprised the dataset of the meta-epidemiological study’

- In the sentence “Most meta-analyses were associated with fairly high or extreme statistical heterogeneity” how did the authors estimate the heterogeneity of the studies? This is not mentioned in the results. Please, consider that I2 cannot estimate the heterogeneity, but I2 represents the percentage of observed variability that is attributed to the variability of true effects.

Authors’ response

We used the restricted maximum likelihood estimator to the between-study heterogeneity parameter, τ2, to estimate the absolute statistical heterogeneity. We also followed the Spiegelhalter et al. to classify the estimated absolute statistical heterogeneity as low, reasonable, fairly high and fairly extreme and this information was mentioned in details in the supplementary materials (Figure S2 in Supplementary Material).

Reference 

Spiegelhalter DJ, Abrams KR, Myles JP. Bayesian approaches to clinical trials and health-care evaluation. John Wiley and Sons, Chichester, 2004.

Discussion

- Almost all results and discussions are misinterpreted. For example: The sentence “The present study found positive association between including non-English studies in orthodontic SRs and SRs with small co-authorship or SRs led by an author affiliated in American institute, but the evidence was inconclusive.” is not accurate. There was no association. Please, revise all the paper and correct it. Also here: “We found that meta-analyses with non-English studies were more likely to yield statistical significance than meta-analyses that included only English studies. However, the evidence was inconclusive due to the small analysed sample”. There was no association. Moreover, for meta-analysis results the authors should present power calculation and discuss it. Probably the results were not associated due to a lack of power and this is related to the limitations that authors impose on papers: 2017 to 2021. So, this results in a limited number of meta-analyses with at least one non-English study. Considering that the authors had two aims, they could increase the period for studies to be included in the meta-analysis to provide an “n” of studies adequate.

Authors’ response

We have not revised the first indicated sentence as follows: ‘The present study found a positive, though, statistically non-significant association between including non-English studies in orthodontic SRs and SRs with small co-authorship or SRs led by an author affiliated in American institute.’

We have also amended the second indicated sentence as follows: ‘We found that meta-analyses with non-English studies were more likely to yield statistical significance than meta-analyses that included only English studies. However, this association was not statistically significant due to the small analyzed sample, which may have led to insufficient power to detect the difference.’

We concur with the Editor about the power issues, associated with the small meta-analyses, which may have led to statistically non-significant associations. However, any post-hoc calculations of power comprise a ‘self-fulfilling prophecy’ because ‘‘post-hoc power is misinterpreted as inadequate power for trials with no statistically significant results, and it does not provide any extra information in the analysis. [...] post-hoc power is a self-fulfilling prophecy that falsely justifies any negative result as a product of a small sample size. Power is defined a priori to determine the sample size needed to estimate a certain effect with a certain type I error. [...] We urge researchers to [...] resort to the vast amount of literature and regulatory guidelines explaining the reasons for avoiding such practice’’; quoted from the article of Christogiannis et al.. As such, we were afraid of justifying false results.

Christogiannis C, Nikolakopoulos S, Pandis N, Mavridis D. The self-fulfilling prophecy of post-hoc power calculations. Am J Orthod Dentofacial Orthop. 2022 Feb;161(2):315-317.

Regarding the number of meta-analyses with at least one non-English study, only 20 studies from the systematic reviews (out of 45) with non-English language included results from these non-English studies in the meta-analysis. Therefore, even if we extend the period by one or two years, it will only marginally increase the number of these meta-analyses, which is unlikely to significantly impact the findings. The primary objective of this study was to illuminate this issue in orthodontic systematic reviews rather than providing an absolute assessment. 

- The limitations of the study must be deeply discussed.

Authors’ response

We have expanded the limitation section to discuss the small included sample and the protocol registration as follows:

‘A wider search is unlikely to change our conclusions on the importance of including non-English studies. However, it may turn statistically non-significant results into statistically significant, particularly when more meta-analyses with non-English language are included in the meta-epidemiological study.’

‘The authors were concerned about the unaddressed correlation that may arise from including more than one meta-analysis from the same SR in the meta epidemiological analysis. As such, only one meta-analysis with non-English study was involved. The prior protocol was not registered due to the constraints imposed by the COVID period, which may introduce some bias. However, the authors diligently adhered to their protocol and rigorous guidelines.’

Figure 3 has a low quality. Please add a high-quality size in Figure 3.

Authors’ response

This was added accordingly.

Review Comments to the Author

Reviewer #1: In general terms, I found the article's proposal interesting, innovative, and well-executed by the authors. Language Bias should be a source of concern within evidence synthesis, given that other studies have already shown how geographical differences can impact submission and publication processes mainly in high-impact journals. However, I believe that some further clarification is needed.

Authors response: We would like to thank the reviewer for the valuable comments that we think will improve our manuscript. In the revised version, we are making efforts to incorporate additional clarification.

Introduction:

Paragraphs 3 and 4 (page 3) present studies that have addressed Language Bias in some way but the definition of Language Bias is unclear in the text. I suggest including a brief definition of Language Bias and the context of English as the dominant scientific spoken/written language in Dentistry.

Authors response: More clarification regarding the language bias detention was added to the introduction according to the reviewer’s comment:

‘The English language is considered the dominant language in research[1] including dentistry. While publications from languages other than English is usually considered as of secondary importance. A previous study [2] found that German authors are more likely to publish trials in English when results are statistically significant, increasing the risk of language bias. Likewise, authors from less developed countries tend to publish more positive results than negative results[3]. Language bias may result from publishing significant findings in English language more than other languages[4]. Subsequently, results from only English language studies could provide a biased assessment of the topic.’ 

Methods:

In the Data Collection Process subsection (page 4): Were there agreement measurements between two reviewers in the pilot assessment of SR?

Authors response: Yes, the pilot data extraction was performed and the conflicts were resolved through a discussion with a third authors until the authors reached 100% agreement. We have added a line to clarify this process:

‘After reaching 100% agreement, the same two authors extracted the data and a third author (SM) cross-checked the collected data.’

Just another small note on the description of the variables collected (page 4 - Data Collection Process subsection): How was the continent variable collected and categorized? Did you consider the affiliation of the first author?

Authors response: Yes, the affiliation of the first author was considered. This was added to the data extraction section:

‘The following characteristics for each SR were extracted: the number of authors, continent of the first author,’

In the Statistical Analysis and Data Synthesis subsection (page 5): In the multivariable binary logistic regression, were all the variables of interest included in the final multivariable regression model? Was there any specific way of selecting variables for the final model? Also, the level of significance (p-value >0.05) could be added here instead in the results section.

Authors response: We did not perform any variable selection; instead, we considered all variables in the multivariable regression model, which is considered optimal statistical practice [1], since all included variables were prespecified (before seeing the data) for being relevant and important to the aims of the study.

We added the following information in the Methods under ‘Multivariable binary logistic regression’:

‘All characteristics were included in the model simultaneously, and no variable selection approach (e.g., stepwise selection) was performed.’

About the significance level, since we report a 95% confidence interval, the significance level is at 5%. We added the following information in that section:

• For the OR results: ‘We concluded a statistically significant association when the 95% CI did not include an OR of 1 (value of no association), which coincides with a p-value below 5% (the significance level); otherwise, the association was statistically non-significant, which coincides with a p-value at least 5%.’

• For the ΔSMD results: ‘We concluded a statistically significant ΔSMD when the 95% CI did not include the value 0, which coincides with a p-value below 5% (the significance level); otherwise, the result was statistically non-significant, which coincides with a p-value at least 5% ‘

[1] Harrell, Frank E., Jr. 2016. Regression Modeling Strategies. Springer Series in Statistics. Cham, Switzerland: Springer International Publishing.

Results:

If possible, present the list of the included systematic reviews as supplementary material to accomplish with the data availability statement.

Authors response: The dataset was shared on open repository: 

https://doi.org/10.5281/zenodo.10201400

Discussion:

In the sentence: “The present study found positive association between including non-English studies in orthodontic SRs and SRs with small co-authorship or SRs led by an author affiliated with an American institute, but the evidence was inconclusive.” (Page 10, 4th paragraph): I believe that an improvement could be made to the wording, considering that "positive association" could be interpreted as a p<0.05, which was not the case in the study. I suggest changing the wording to "a relationship between ..." or some similar term at the authors' discretion.

Authors response: this sentence was amended according to the editors’ and reviewer’s comment, as follows:

‘The present study found a positive, though, statistically non-significant association between including non-English studies in orthodontic SRs and SRs with small co-authorship or SRs led by an author affiliated in American institute.’

We have included the term ‘statistically non-significant association’ to avoid a misinterpretation like ‘positive association" could be interpreted as a p<0.05’. Whether an association (measured using an OR) is positive or negative does *not* dependent on whether a p-value is below or above a significance threshold; it dependents on whether the OR is above or below 1 (value of no association), respectively. Whether an association (which is another term for relationship) is statistically significant (conclusive evidence) or statistically non-significant (inconclusive evidence) dependents on whether a p-value is below or above a significance threshold, respectively.

Regarding limitations and strength subsection: In the metanalysis, could the effect measures found be affected by the heterogeneity of outcomes considered as the main outcome? Could this be considered a limitation of this study?

Authors response: The limitations was amended. According to our understanding, the reviewer is asking for the effect of heterogeneity in the included meta-analysis? We have used the standardized mean difference due to the variation in our sample to minimize these effects. However, the statical heterogeneity was in the individual meta-analyses and not in our meta epidemiological analysis. 

Also, we have reported the following: Most meta-analyses were associated with fairly high or extreme statistical heterogeneity, regardless of language restriction (S2 Figure). Restricting inclusion to English studies increased statistical heterogeneity in almost half meta-analyses compared to including all studies regardless of language (S2 Figure).

If the reviewer can provide more details, we are happy to follow his/her guidance. 

Reviewer #2: Dear Editor Luiz Alexandre Chisini,

Thank you for the opportunity to review this article.

I hope you find my feedback valuable.

Major Review

This article aims to assess language bias in systematic reviews published in orthodontics;

however, it may incorporate other biases at various levels:

Authors response: We would like to thank the reviewer for the valuable comments that we think will improve our manuscript.

Searches were conducted only in two databases. While I understand that the majority

of relevant literature might be present in these databases, this choice needs a more

robust justification. A sample calculation can be used to justify the number of articles that need to be included, potentially reducing the need for searches in additional databases.

There is also a significant date restriction for the search that needs clarification

Authors response: We agree with the reviewer regarding these concerns. This was discussed between the authors before conducting the search due to its importance. Our understanding was to include the largest number of published systematic reviews in the five leading orthodontic journals for the last five years. The search in PubMed was undertaken to retrieve systematic reviews published in these journals which is indexed in PubMed. While Oral health group was searched to retrieve the relevant Cochrane reviews.

Meta epidemiological study aim to investigate whether the characteristic of interest influences the treatment effect[5]. It a common practice in meta-epidemiological research to focus on specific journals in a specific period, and this is always incorporated in the aim of the study. For instance, these meta epidemiological studies[6-8] were published in a prestigious orthodontic journal, and this study[9] was published in BMJ with one year search. Likewise, published meta-epidemiological studies [10, 11] in journal of clinical epidemiology have a restricted search. 

We have amended the following information in the Methods under Search and study selection: One author (SM) performed the search of PubMed using text words and medical subject headings to retrieve systematic reviews published in the leading orthodontic journals indexed in PubMed (Table1). All relevant Cochrane orthodontic reviews within the same period were also retrieved by another author (BD) through the Cochrane Oral Health Group.

The most critical point is that there has been no prior publication of the study protocol.

Authors’ response 

There was a prior protocol to ensure a well-conducted study. Unfortunately, the authors did not register it on PROSPERO due to the constraints imposed by the COVID period and the simultaneous need for prompt initiation of the study. We have included this point in the study limitations. 

We have added the following information in the limitations:

‘The prior protocol was not registered due to the constraints imposed by the COVID period, which may introduce some bias. However, the authors diligently adhered to their protocol and rigorous guidelines.’

Minnor Review

Other minor points that also require attention:

It would be beneficial to make it clearer what the primary outcome of this study is.

Authors response A clarification was added to the methods according to the reviewer’s comment. “Specifically, we considered the following ‘algorithm’: if the meta-analysis of primary outcome included at least three studies with at least one being non-English, this meta-analysis was included in our collection. If more meta-analyses were eligible, we opted for the first one addressing the primary outcome. If the meta-analysis of the primary outcome was not eligible, the meta-analysis for secondary outcomes was checked.”

From what I understand, the analysis considered studies that included both non-English language articles and those that did not. In the results section, the authors report that there were no explicit language restrictions ( = 135, 80%) or explicit exclusion of non-English studies ( = 170, 98%) in the majority of systematic reviews. However, there is a possibility that some systematic reviews may include studies in other languages, even if these studies do not exist and, consequently, are not found. How would they be considered in the analysis? Perhaps, this potential scenario could be addressed in the discussion.

Authors response: in the rare case, the authors may restrict their search to English studies to minimize the results of the search that may facilitate the screening of the potential records. However, some authors perform a manual search to handle the gray literature and unregistered studies as well as studies through google scholar search. This may result in a non-English study which was may happen in one[12] of our included sample. This was added to the discussion:

‘In this regard, manual search, particularly in google scholar may result in non-English studies even if the search was restricted in the other bibliographies. This was evident in one included SR [25] with additional manual search, although the search was restricted in the searching bibliographies. ’

In the results section, when indicating the p-value, please provide the exact number rather than using p-value > 0.05. If the p-value is very small, describe it as p-value < 0.0001.

Authors response:

A p-value < 0.05 coincides with a 95% confidence interval that does not include the value of no association (i.e., 1 for OR, and 0 for ΔSMD), and a p-value at least 5% coincides with a 95% confidence interval that includes the value of no association. Therefore, in the Results, we emphasize on the 95% confidence intervals for conveying more information (about the range of values and the statistical significance) for interpretation than a p-value (it only informs about statistical significance).

We have also included the following information in the Methods:

• For the OR results: ‘We concluded a statistically significant association when the 95% CI did not include an OR of 1 (value of no association), which coincides with a p-value below 5% (the significance level); otherwise, the association was statistically non-significant, which coincides with a p-value at least 5%. ’

• For the ΔSMD results: ‘We concluded a statistically significant ΔSMD when the 95% CI did not include the value 0, which coincides with a p-value below 5% (the significance level); otherwise, the result was statistically non-significant, which coincides with a p-value at least 5% ‘

I also suggest including the "numerical" values in the results presented in the abstract.

Authors response: this has been amended accordingly.

Please define what would constitute an inconclusive association. Was there no

association? What is the power of the test used to prevent Type 1 error?

Authors response: We have replaced ‘conclusive evidence’ with ‘statistically significant association’ and ‘weak or inconclusive evidence’ with ‘statistically non-significant association’.

We understand that the reviewer is interested in the power calculation, which is a post-hoc calculation in this case. However, any post-hoc calculations of power comprise a ‘self-fulfilling prophecy’ because ‘‘post-hoc power is misinterpreted as inadequate power for trials with nonstatistically significant results, and it does not provide any extra information in the analysis. [...] Post-hoc power is a self-fulfilling prophecy that falsely justifies any negative result as a product of a small sample size. Power is defined a priori to determine the sample size needed to estimate a certain effect with a certain type I error. [...] We urge researchers to [...] resort to the vast amount of literature and regulatory guidelines explaining the reasons for avoiding such practice’’; quoted from the article of Christogiannis et al. and we concur with the authors of the article.

Christogiannis C, Nikolakopoulos S, Pandis N, Mavridis D. The self-fulfilling prophecy of post-hoc power calculations. Am J Orthod Dentofacial Orthop. 2022 Feb;161(2):315-317.

In the discussion, the publication bias can be explored. As the authors themselves point

out, studies in other languages may exhibit lower methodological quality.

Authors response: Investigating publication bias in orthodontic is challenging due to the low number of included studies (usually <10 studies)[13], and it will not differentiate between the included English and non-English studies. As this make the interpretation of publication bias very difficult, we have recommended a sensitivity analysis to do so. The following information can be found in the Discussion 

‘Moreover, different studies [24, 31-33] assessed the quality of the English and non-English studies, and found a higher risk of bias in non-English studies due to suboptimal randomization, insufficient reporting of the blinding, and incomplete data. As such, non-English studies may be removed from the meta-analysis in the context of a sensitivity analysis to inspect the robustness of the meta-analysis results.’

Additionally, it is essential to provide the search strategy used for the Cochrane Library

database.

Authors response: The search was undertaken manually through the Cochrane Oral Health Group.

The theme of this study is very interesting. With some corrections to strengthen the methodological robustness, there is a good potential for publication.

Authors response: We would like to thank the reviewer again for his valuable comments that improved our understanding for the topic and our study limitations.

References

1. FERGUSON G, PÉREZ-LLANTADA C, PLO R. English as an international language of scientific publication: a study of attitudes. World Englishes. 2011;30(1):41-59. doi: https://doi.org/10.1111/j.1467-971X.2010.01656.x.

2. Egger M, Zellweger-Zähner T, Schneider M, Junker C, Lengeler C, Antes G. Language bias in randomised controlled trials published in English and German. The Lancet. 1997;350(9074):326-9. doi: 10.1016/s0140-6736(97)02419-7.

3. Panagiotou OA, Contopoulos-Ioannidis DG, Ioannidis JP. Comparative effect sizes in randomised trials from less developed and more developed countries: meta-epidemiological assessment. BMJ. 2013;346:f707. Epub 20130212. doi: 10.1136/bmj.f707. PubMed PMID: 23403829; PubMed Central PMCID: PMCPMC3570069.

4. J B, EA S, C H. Langauge bias. In: Catalogue Of Bias 2017. Available from: https://www.catalogueofbias.org/biases/language-bias.

5. Kataoka Y, Banno M, Tsujimoto Y, Furukawa TA. "Meta-epidemiological study" is a study in which the unit of analysis is a study, not a patient; response to Puljak et al. J Clin Epidemiol. 2023;154:219-20. Epub 20221209. doi: 10.1016/j.jclinepi.2022.12.002. PubMed PMID: 36503003.

6. Tatas Z, Koutsiouroumpa O, Seehra J, Mavridis D, Pandis N. Do pooled estimates from orthodontic meta-analyses change depending on the meta-analysis approach? A meta-epidemiological study. Eur J Orthod. 2023. Epub 20230712. doi: 10.1093/ejo/cjad031. PubMed PMID: 37435902.

7. Mheissen S, Khan H, Seehra J, Pandis N. Are longitudinal randomised controlled oral health trials properly analysed? A meta-epidemiological study. Journal of dentistry. 2022;124:104182. Epub 20220609. doi: 10.1016/j.jdent.2022.104182. PubMed PMID: 35691454.

8. Mheissen S, Khan H, Almuzian M, Alzoubi EE, Pandis N. Do longitudinal orthodontic trials use appropriate statistical analyses? A meta-epidemiological study. Eur J Orthod. 2021. Epub 2021/09/26. doi: 10.1093/ejo/cjab069. PubMed PMID: 34561710.

9. Moustgaard H, Clayton GL, Jones HE, Boutron I, Jorgensen L, Laursen DRT, et al. Impact of blinding on estimated treatment effects in randomised clinical trials: meta-epidemiological study. BMJ. 2020;368:l6802. Epub 2020/01/23. doi: 10.1136/bmj.l6802. PubMed PMID: 31964641; PubMed Central PMCID: PMCPMC7190062 at www.icmje.org/coi_disclosure.pdf (available on request from the corresponding author) and declare: no support from any organisation for the submitted work; no financial relationships with any organisations that might have an interest in the submitted work in the previous three years; no other relationships or activities that could appear to have influenced the submitted work.

10. Tsujimoto Y, Tsujimoto H, Kataoka Y, Kimachi M, Shimizu S, Ikenoue T, et al. Majority of systematic reviews published in high-impact journals neglected to register the protocols: a meta-epidemiological study. J Clin Epidemiol. 2017;84:54-60. Epub 20170227. doi: 10.1016/j.jclinepi.2017.02.008. PubMed PMID: 28242481.

11. Smail-Faugeron V, Tan A, Caille A, Yordanov Y, Hajage D, Tubach F, et al. Meta-analyses frequently include old trials that are associated with a larger intervention effect: a meta-epidemiological study. Journal of Clinical Epidemiology. 2022;145:144-53. doi: https://doi.org/10.1016/j.jclinepi.2022.01.023.

12. Elsten E, Caron C, Dunaway DJ, Padwa BL, Forrest C, Koudstaal MJ. Dental anomalies in craniofacial microsomia: A systematic review. Orthod Craniofac Res. 2020;23(1):16-26. Epub 20191028. doi: 10.1111/ocr.12351. PubMed PMID: 31608577; PubMed Central PMCID: PMCPMC7003932.

13. Koletsi D, Fleming PS, Eliades T, Pandis N. The evidence from systematic reviews and meta-analyses published in orthodontic literature. Where do we stand? Eur J Orthod. 2015;37(6):603-9. Epub 2015/02/11. doi: 10.1093/ejo/cju087. PubMed PMID: 25667037.

---

## [Decision Letter · Decision Letter 1]

7 Feb 2024

PONE-D-23-28166R1Language Bias in Orthodontic Systematic Reviews: A Meta-epidemiological StudyPLOS ONE

Dear Dr. Mheissen,

Thank you for submitting your manuscript to PLOS ONE. After careful consideration, we feel that it has merit but does not fully meet PLOS ONE’s publication criteria as it currently stands. Therefore, we invite you to submit a revised version of the manuscript that addresses the points raised during the review process.

We look forward to receiving your revised manuscript.

Kind regards,

Luiz Alexandre Chisini, Ph.D

Academic Editor

PLOS ONE

**Additional Editor Comments:**

I have still concerns about the paper. The paper's objective is very interesting. However, the interpretation of the results is still incorrect. The authors remain doing affirmations that cannot be maintained by the statistics. So, the paper spin is very high and can lead the authors to misinterpretation.

- When the authors mention “174 SRs were eligible for inclusion in this study” it is not possible to undertint if all 174 were included or not for the revision. They are eligible for inclusion, but could not be included. So, it is better to readers understand the number of “included” instead of “eligible”. This is confusing in the abstract and in the results. Please, Make clear the number of articles included in the review and meta-analysis at the beginning of the results sections (in the abstract and results)

- Please use the exact number of p-value.

- I have also concerns about the sentence: “As such, the evidence of the overestimation of meta-analysis results with non-English studies is inconclusive”. The meta-analysis with random models (which is the right model for studies with methodological variation) shows no difference. So, could be “inconclusive” only isn’t power enough, but then there are problems with the sample size. On the other hand, I believe that the conclusion is there was no evidence of significant overestimation.

- I dindnt found the result for this sentence: “The univariate binary logistic regression indicated that the odds of statistical significance in the summary effect estimate increased by 220% (OR: 3.20) in systematic reviews with non-English studies than in systematic reviews with only English studies. However, the evidence was weak because the 95% CI (0.57, 18.92) was substantially wide and included the null value”. Also, the authors cannot affirm that increase, because the 95% CI doesn’t allow this. Please, revise affirmations that cannot be supported by statistics.

- Here, another misinformation: “The present study found a positive, though, statistically non-significant association between including non-English studies in orthodontic SRs and SRs with small coauthorship or SRs led by an author affiliated in American institute”. The authors cannot affirm that the is a “positive” but non-significant association. Or there is an association or there is not. Please, revise.

- In the sentence “Our meta-epidemiological analysis demonstrated a larger summary effect size from including non-English studies, though the evidence was inconclusive due to material statistical heterogeneity”. Here again, they are not “inconclusive”.

- This sentence “A wider search is unlikely to change our conclusions on the importance of including non-English studies.” Is a supposition and cannot be maintained with data. So, I recommend the authors to avoid this type of affirmation.

- Both in the conclusion as the abstract conclusion, the authors cannot support this affirmation “Language restriction seems to have a non-negligible impact on the results of orthodontic SRs.” There was no change in the results according to the results. This interpretation leads to a spin on the study.

In Table 2, please, standardize the reference category as the first row

In table 3, please, provide the p-value

Reviewers' comments:

Reviewer's Responses to Questions

**Comments to the Author**

1. If the authors have adequately addressed your comments raised in a previous round of review and you feel that this manuscript is now acceptable for publication, you may indicate that here to bypass the “Comments to the Author” section, enter your conflict of interest statement in the “Confidential to Editor” section, and submit your "Accept" recommendation.

Reviewer #1: All comments have been addressed

Reviewer #2: All comments have been addressed

2. Is the manuscript technically sound, and do the data support the conclusions?

Reviewer #1: Yes

Reviewer #2: Yes

3. Has the statistical analysis been performed appropriately and rigorously? 

Reviewer #1: Yes

Reviewer #2: I Don't Know

4. Have the authors made all data underlying the findings in their manuscript fully available?

Reviewer #1: Yes

Reviewer #2: Yes

5. Is the manuscript presented in an intelligible fashion and written in standard English?

Reviewer #1: Yes

Reviewer #2: Yes

6. Review Comments to the Author

Reviewer #1: I would like to thank you for the opportunity to review this article again, as well as for the authors' thoughtful response. The authors have made considerable improvements to the paper that have enhanced the way the findings are communicated. Below are some minor corrections that can be made to ensure the best version of this manuscript.

Abstract:

In the "Eligibility Criteria" section, it is clearly stated that the two main criteria for selecting SRs for inclusion were (1) publication between 2017 and 2021 and (2) being published in one of the five highest-impact journals indexed in PubMed, in addition to Cochrane reviews from the same period. In this way, I believe that in the "Data Source" section of the Abstract, the wording could be improved by inserting separately that the SRs were retrieved from five high-impact journals indexed in PubMed. After this, could be inserted that searches were carried out in the Cochrane database, also considering the publication period (2017-2021).

Methods:

In the response letter (page 8), the authors stated that in the Statistical Analysis and Data Synthesis subsection, two sentences had been included to address the issue of statistical significance interpretation in different analyses carried out. However, the second of these, copied below, was not present in the main text. Please revise the insertion of the information.

“•For the ΔSMD results: ‘We concluded a statistically significant ΔSMD when the 95% CI did not include the value 0, which coincides with a p-value below 5% (the significance level); otherwise, the result was statistically non-significant, which coincides with a p-value at least 5%”

Results:

Please verify Table S1: Although the Systematic review selection subsection mentions 11 excluded studies with reasons, only 10 titles are shown in Table S1.

Reviewer #2: Most of my suggestions were accepted. The ones that couldn't be altered were justified accordingly. I appreciate the opportunity to review this study and recommend it for publication. Additionally, I would like to note that the authors have introduced an interesting and underexplored topic in the scientific literature.

7. PLOS authors have the option to publish the peer review history of their article (what does this mean?). If published, this will include your full peer review and any attached files.

Reviewer #1: **Yes: **Letícia Regina Morello Sartori

Reviewer #2: No

---

## [Author Response · Author response to Decision Letter 1]

8 Feb 2024

Dear editor Luiz Alexandre Chisini, 

We would like to thank you and the reviewers for the comments and suggestions that we think will improve our manuscript. 

Please find below responses and actions taken. In the revised manuscript we highlight amended sections.

Additional Editor Comments:

I have still concerns about the paper. The paper's objective is very interesting. However, the interpretation of the results is still incorrect. The authors remain doing affirmations that cannot be maintained by the statistics. So, the paper spin is very high and can lead the authors to misinterpretation.

- When the authors mention “174 SRs were eligible for inclusion in this study” it is not possible to undertint if all 174 were included or not for the revision. They are eligible for inclusion, but could not be included. So, it is better to readers understand the number of “included” instead of “eligible”. This is confusing in the abstract and in the results. Please, Make clear the number of articles included in the review and meta-analysis at the beginning of the results sections (in the abstract and results)

Authors’ response: the results and the abstract were amended according to the Editors’ comments, as follows:

Abstract (Results): ‘174 SRs were included in this study.’

Results (Systematic review selection): ‘After full text reading of one hundred eighty-five SRs, 174 SRs were included in the present study.’

- Please use the exact number of p-value.

Authors’ response: We have now added the p-value for the random-effects and fixed-effect model in the Results subsection ‘Examining the influence of non-English studies on summary results’. 

- I have also concerns about the sentence: “As such, the evidence of the overestimation of meta-analysis results with non-English studies is inconclusive”. The meta-analysis with random models (which is the right model for studies with methodological variation) shows no difference. So, could be “inconclusive” only isn’t power enough, but then there are problems with the sample size. On the other hand, I believe that the conclusion is there was no evidence of significant overestimation.

Authors’ response: We amended the indicated sentence as follows: ‘As such, the overestimation of meta-analysis results by including non-English studies was statistically non-significant’.

- I dindnt found the result for this sentence: “The univariate binary logistic regression indicated that the odds of statistical significance in the summary effect estimate increased by 220% (OR: 3.20) in systematic reviews with non-English studies than in systematic reviews with only English studies. However, the evidence was weak because the 95% CI (0.57, 18.92) was substantially wide and included the null value”. Also, the authors cannot affirm that increase, because the 95% CI doesn’t allow this. Please, revise affirmations that cannot be supported by statistics.

Authors’ response: To avoid confusing the readers, we amended the indicated sentence as follows:

‘The univariate binary logistic regression indicated that the odds of statistical significance in the summary effect estimate was 3.20 times larger in systematic reviews with non-English studies than in systematic reviews with only English studies. However, the association was statistically non-significant (OR:3.20, 95%CI: 0.57, 18.92, P=0.18).’

- Here, another misinformation: “The present study found a positive, though, statistically non-significant association between including non-English studies in orthodontic SRs and SRs with small coauthorship or SRs led by an author affiliated in American institute”. The authors cannot affirm that the is a “positive” but non-significant association. Or there is an association or there is not. Please, revise.

Authors’ response: We removed ‘positive, though,’ from the indicated sentence.

- In the sentence “Our meta-epidemiological analysis demonstrated a larger summary effect size from including non-English studies, though the evidence was inconclusive due to material statistical heterogeneity”. Here again, they are not “inconclusive”.

Authors’ response: We replaced ‘the evidence was inconclusive’ with ‘this was not statistically significant’.

- This sentence “A wider search is unlikely to change our conclusions on the importance of including non-English studies.” Is a supposition and cannot be maintained with data. So, I recommend the authors to avoid this type of affirmation.

Authors’ response: We amended the indicated sentence as follows:

‘A wider search may have some impact on the importance of including non-English studies.’

- Both in the conclusion as the abstract conclusion, the authors cannot support this affirmation “Language restriction seems to have a non-negligible impact on the results of orthodontic SRs.” There was no change in the results according to the results. This interpretation leads to a spin on the study.

Authors’ response: Professors Doug Altman and Martin Bland wrote a seminar commentary to raise awareness on how results from statistical tests should be interpreted. The article’s title is ‘Absence of evidence is not evidence of absence’ (1); namely, finding a p-value above the selected level of significance (e.g., 5%), and thus, failing to reject the null hypothesis does not mean that there is no difference in the compared groups. It just means that the difference is statistically non-significant, and hence, we cannot draw any firm conclusion in favour or against any of the compared groups due to lack of sufficient information (e.g., small sample size) or substantial variation in the measurements, or a combination of both. Taken from the abstract of Altman and Bland: “When statistical analysis of the study data finds a P value greater than 5%, it is convention to deem the assessed difference nonsignificant. Just because convention dictates that such study findings be termed nonsignificant, or negative, however, it does not necessarily follow that the study found nothing of clinical importance.” (1). And lastly, from the last paragraph of the article: “When we are told that “there is no evidence that A causes B” we should first ask whether absence of evidence means simply that there is no information at all. If there are data we should look for quantification of the association rather than just a P value. Where risks are small P values may well mislead: confidence intervals are likely to be wide, indicating considerable uncertainty. While we can never prove the absence of a relation, when necessary we should seek evidence against the link between A and B—for example, from case-control studies.” (1)

We amended the indicated sentence as follows: ‘Language restriction seems to have no statistically significant impact on the results of orthodontic SRs’

Reference

(1) Altman DG, Bland JM. Absence of evidence is not evidence of absence. BMJ. 1995 Aug 19;311(7003):485. doi: 10.1136/bmj.311.7003.485.

In Table 2, please, standardize the reference category as the first row

In table 3, please, provide the p-value

Authors’ response: Tables were amended accordingly.

Reviewers' comments:

Reviewer's Responses to Questions

Comments to the Author

Reviewer #1: I would like to thank you for the opportunity to review this article again, as well as for the authors' thoughtful response. The authors have made considerable improvements to the paper that have enhanced the way the findings are communicated. Below are some minor corrections that can be made to ensure the best version of this manuscript.

Authors response: We would like to thank the reviewer for the valuable comments that have improved our manuscript. In the revised version, we are making efforts to incorporate additional clarification.

Abstract:

In the "Eligibility Criteria" section, it is clearly stated that the two main criteria for selecting SRs for inclusion were (1) publication between 2017 and 2021 and (2) being published in one of the five highest-impact journals indexed in PubMed, in addition to Cochrane reviews from the same period. In this way, I believe that in the "Data Source" section of the Abstract, the wording could be improved by inserting separately that the SRs were retrieved from five high-impact journals indexed in PubMed. After this, could be inserted that searches were carried out in the Cochrane database, also considering the publication period (2017-2021).

Authors’ response: We have amended the Data source section in Abstract as follows:

‘SRs published in high-impact orthodontic journals between 2017 and 2021 were retrieved through an electronic search of PubMed. Additionally, Cochrane oral health group was searched in June 2022 for orthodontic systematic reviews published in the same period.’

Methods:

In the response letter (page 8), the authors stated that in the Statistical Analysis and Data Synthesis subsection, two sentences had been included to address the issue of statistical significance interpretation in different analyses carried out. However, the second of these, copied below, was not present in the main text. Please revise the insertion of the information.

“•For the ΔSMD results: ‘We concluded a statistically significant ΔSMD when the 95% CI did not include the value 0, which coincides with a p-value below 5% (the significance level); otherwise, the result was statistically non-significant, which coincides with a p-value at least 5%”

Authors’ response: Thank you for this observation. We have now added all P values to the text for better clarity. However, if the reviewer thinks that we should add this sentence, we will be happy to follow his/her guidance.

Results:

Please verify Table S1: Although the Systematic review selection subsection mentions 11 excluded studies with reasons, only 10 titles are shown in Table S1.

Authors’ response: Sorry for this mistake. This excluded study was added to the table.

Reviewer #2: Most of my suggestions were accepted. The ones that couldn't be altered were justified accordingly. I appreciate the opportunity to review this study and recommend it for publication. Additionally, I would like to note that the authors have introduced an interesting and underexplored topic in the scientific literature.

Authors response: We would like to thank the reviewer for the valuable comments that have improved our manuscript.

---

## [Decision Letter · Decision Letter 2]

7 Mar 2024

Language Bias in Orthodontic Systematic Reviews: A Meta-epidemiological Study

PONE-D-23-28166R2

Dear Dr. Samer Mheisse

We’re pleased to inform you that your manuscript has been judged scientifically suitable for publication and will be formally accepted for publication once it meets all outstanding technical requirements.

Kind regards,

Luiz Alexandre Chisini, Ph.D

Academic Editor

PLOS ONE

Additional Editor Comments (optional):

Reviewers' comments:

Reviewer's Responses to Questions

**Comments to the Author**

1. If the authors have adequately addressed your comments raised in a previous round of review and you feel that this manuscript is now acceptable for publication, you may indicate that here to bypass the “Comments to the Author” section, enter your conflict of interest statement in the “Confidential to Editor” section, and submit your "Accept" recommendation.

Reviewer #1: All comments have been addressed

2. Is the manuscript technically sound, and do the data support the conclusions?

Reviewer #1: Yes

3. Has the statistical analysis been performed appropriately and rigorously? 

Reviewer #1: Yes

4. Have the authors made all data underlying the findings in their manuscript fully available?

Reviewer #1: Yes

5. Is the manuscript presented in an intelligible fashion and written in standard English?

Reviewer #1: Yes

6. Review Comments to the Author

Reviewer #1: Dear authors, my concerns about the article have been successfully solved.

Thank you very much for your careful review.

7. PLOS authors have the option to publish the peer review history of their article (what does this mean?). If published, this will include your full peer review and any attached files.

Reviewer #1: **Yes: **Letícia Sartori
